# Biological invasions facilitate zoonotic disease emergences

Lin Zhang[1,2,11], Jason Rohr [ORCID][3,11], Ruina Cui[1,11], Yusi Xin[4], Lixia Han[5,6], Xiaona Yang[7], Shimin Gu[1], Yuanbao Du [ORCID][1], Jing Liang[8], Xuyu Wang[1,9], Zhengjun Wu[5,6], Qin Hao[2✉] & Xuan Liu [ORCID][1,10,11✉]

Outbreaks of zoonotic diseases are accelerating at an unprecedented rate in the current era of globalization, with substantial impacts on the global economy, public health, and sustainability. Alien species invasions have been hypothesized to be important to zoonotic diseases by introducing both existing and novel pathogens to invaded ranges. However, few studies have evaluated the generality of alien species facilitating zoonoses across multiple host and parasite taxa worldwide. Here, we simultaneously quantify the role of 795 established alien hosts on the 10,473 zoonosis events across the globe since the 14th century. We observe an average of ~5.9 zoonoses per alien zoonotic host. After accounting for species-, disease-, and geographic-level sampling biases, spatial autocorrelation, and the lack of independence of zoonosis events, we find that the number of zoonosis events increase with the richness of alien zoonotic hosts, both across space and through time. We also detect positive associations between the number of zoonosis events per unit space and climate change, land-use change, biodiversity loss, human population density, and PubMed citations. These findings suggest that alien host introductions have likely contributed to zoonosis emergences throughout recent history and that minimizing future zoonotic host species introductions could have global health benefits.

[1] Key Laboratory of Animal Ecology and Conservation Biology, Institute of Zoology, Chinese Academy of Sciences, 1 Beichen West Road, Chaoyang, 100101 Beijing, China. [2] State Key Laboratory for Infectious Disease Prevention and Control, National Institute for Communicable Disease Control and Prevention, Chinese Center for Disease Control and Prevention, Changping, 102206 Beijing, China. [3] Department of Biological Sciences, Environmental Change Initiative, University of Notre Dame, Notre Dame, IN 46556, USA. [4] School of Landscape and Architecture, Beijing Forestry University, Haidian, 100083 Beijing, China. [5] Key Laboratory of Ecology of Rare and Endangered Species and Environmental Protection (Guangxi Normal University), Ministry of Education, Guilin 541004, China. [6] Guangxi Key Laboratory of Rare and Endangered Animal Ecology, College of Life Science, Guangxi Normal University, Guilin 541004, China. [7] Daxing Center for Disease Control and Prevention, Daxing, 102600 Beijing, China. [8] College of Veterinary Medicine, Sichuan Agricultural University, Chengdu 611130, China. [9] Institute of Physical Science and Information Technology, Anhui University, Hefei 230601, China. [10] University of Chinese Academy of Sciences, 100049 Beijing, China. [11] These authors contributed equally: Lin Zhang, Jason Rohr, Ruina Cui, Xuan Liu. ✉email: haoqin@icdc.cn; liuxuan@ioz.ac.cn

The rapid increase in zoonotic diseases (i.e., diseases caused by pathogens that are spread from animal to people, also known as zoonoses) poses a significant threat to the global economy, public health, and social stability[1,2]. Positive associations between alien animal host invasions and the incidence of zoonosis has long been of concern[3–6], especially given recent increases in both alien species introductions[7] and zoonoses, such as avian influenza, SARS, and COVID-19[8–11]. Established alien species can contribute to zoonosis by increasing the abundance of existing pathogens and introducing novel pathogens[12,13]. In addition, most alien animal introductions are associated with human activities such as pet trade and aquaculture that may provide more contact opportunities between alien hosts and humans[8]. In recent years, there have been increasing reports linking alien host species with zoonoses. For example, invasive rats have contributed to the emergence of plague, murine typhus, scrub typhus, leptospirosis, and hantavirus hemorrhagic fever throughout the world[14], introduced North American raccoons contributed to the emergence of West Nile virus and human roundworm infections in Europe[15], the spread of alien mosquitos in Europe contributed to the emergence of chikungunya and dengue fever[16], and alien lice and flea vectors have fomented epidemics of typhus and plague in established ranges[13]. Although these individual examples provide striking geographical and taxonomic evidence of the transmission of zoonoses by alien hosts[6], there has not previously been a global synthesis of the role of established alien zoonotic hosts on zoonoses across a broad range of taxonomic groups.

Animal invasions and zoonosis emergences are often correlated with many cofactors, such as propagule pressure (a composite measure of the number of individuals released into a region)[17]; other global change factors, such as climate change[18], biodiversity loss[19], and land-use change[20]; human population density[1]; and disease and invasive species surveillance and research efforts[21]. However, few studies at the global scale, across host and parasite taxa, have controlled for these various cofactors to better isolate the unique contribution of established alien host species to zoonoses, hindering the development of effective regulations for alien species introductions and public health.

To address this knowledge gap, we use a global database of 10,473 zoonosis events since the year 1348 (compiled with the assistance of GIDEON)[22] and evaluate the role of established alien zoonotic hosts on the number of zoonosis events across the globe, controlling for propagule pressure factors (non-zoonotic host introductions and human population density), climate (global environmental stratification, GenS), global change processes (climate change, land-use modification and biodiversity loss), native biodiversity, sampling effort (country surveillance capacity and reporting bias), and spatial autocorrelation (longitude and latitude of the geographic centroid of each administrative area) (Fig. 1). Importantly, a correlation between zoonosis events and zoonotic alien hosts might just be a product of areas with high propagule pressure having more introductions of both hosts and pathogens[13,17]. To address this potential issue, we first conduct an intensive literature review to identify each established alien species as a zoonotic or non-zoonotic host species. We then include non-zoonotic alien host richness as a positive control for propagule pressure that cannot directly increase zoonosis emergences. Thus, a significant effect of zoonotic alien host introductions on the number of zoonosis events when controlling for the number of non-zoonotic alien host introductions (and other covariates) would provide an estimate of the causal effect of established zoonotic alien host species on zoonosis independent of propagule pressure. Given that taxonomic groupings within classes of hosts can vary in contributions to zoonoses[6,23–25], we also conduct additional analyses to identify which alien taxonomic groups seem to contribute most to past zoonosis events. Moreover, a correlation between alien species invasions and zoonosis events in space may be spurious if the disease

occurred earlier than the alien animal arrivals. Therefore, we also conduct analyses to explore the relationship between species invasions and zoonosis events through time. Finally, we generate a global map showing where historical zoonoses were likely most influenced by alien host introductions, which might provide helpful insights into our understanding on the potential effects of alien animal invasions on future zoonosis emergences.

Here, after controlling for other variables, we show that number of zoonosis events increase with the richness of alien zoonotic hosts across mammalian (particularly three orders: Artiodactyla, Carnivora, and Rodentia), birds (particularly waterfowl, Galliformes, and Passeriformes), and Dipteran invertebrate host species both across space and through time. Importantly, we do not observe a significant effect of the alien non-zoonotic host species, indicating that our findings on the correlation between the number of zoonosis events and the number of zoonotic alien hosts are unlikely to be a byproduct of areas with high propagule pressure having more introductions of both hosts and pathogens.

## Results

**Global patterns of historical zoonoses and established alien zoonotic hosts.** As alien species may carry both existing and novel pathogens and transmit them to local native species that can transmit them to humans[6], our analyses included events caused by both re-emerging and novel zoonoses in each of a total of 201 countries or regions worldwide (Supplementary Data 1). Of the 10,473 events for 161 zoonoses reported from 1348 to 2020, 2970 events were caused by emerging diseases introduced to a new administrative unit and 7503 events were caused by re-emerging diseases that had been reported before. Overall, these zoonoses were most often associated with mammalian hosts, followed by avian, invertebrate, and herpetofaunal hosts (Supplementary Fig. 1A, Supplementary Data 2–4, note that some diseases may be carried by multiple host taxa). To calculate the number of zoonosis carried by the alien zoonotic hosts, we first obtained a total of 93,544 pathogen-alien host records from an intensive literature and database review (Supplementary Data 2). Using these records, we determined that at least 35.6% (283/795) of established alien animals are zoonotic hosts of one or more zoonoses and there is an average of 5.9 (±0.58, average ±S.E.) zoonoses per alien zoonotic host (Supplementary Data 3). The number of zoonoses varied among alien host taxa (Fig. 2), which was predominated by alien mammalian zoonotic hosts, especially from orders Artiodactyla (17.5 ± 3.70), Carnivora (12.2 ± 3.13), and Rodentia (8.9 ± 1.97), and by alien avian hosts, especially from waterfowl (4.1 ± 0.78), Galliformes (3.8 ± 1.05), and Passeriformes (2.5 ± 0.36), and by alien Dipterans (5.0 ± 0.58) for invertebrate-hosted zoonoses (Fig. 2).

These zoonosis events most often occurred in Europe (3495 events), followed by Asia (2180), North America (2120), Africa (1301), South America (897) and Oceania (480, Supplementary Fig. 1B). In general, these zoonosis events were concentrated in higher latitudes, such as the United States and Western Europe (Supplementary Fig. 2). Across pathogens, zoonoses were largely caused by bacteria (4622) and viruses (4002), followed by parasitic animals (1621) and fungi (225, Supplementary Fig. 1B).

**Spatial relationship between alien animal invasions and zoonotic diseases.** We next used generalized additive mixed models (GAMMs) to examine the relationship between the number of zoonotic alien host species and zoonosis events in a region controlling for sampling efforts at species-, disease - and geographic-levels, administrative region size, alien non-zoonotic host introductions (i.e., propagule pressure control), and various other covariates, while treating continent (i.e., human settlement history), host order, and pathogen identity as random intercepts (to control for the lack of independence associated with these factors;

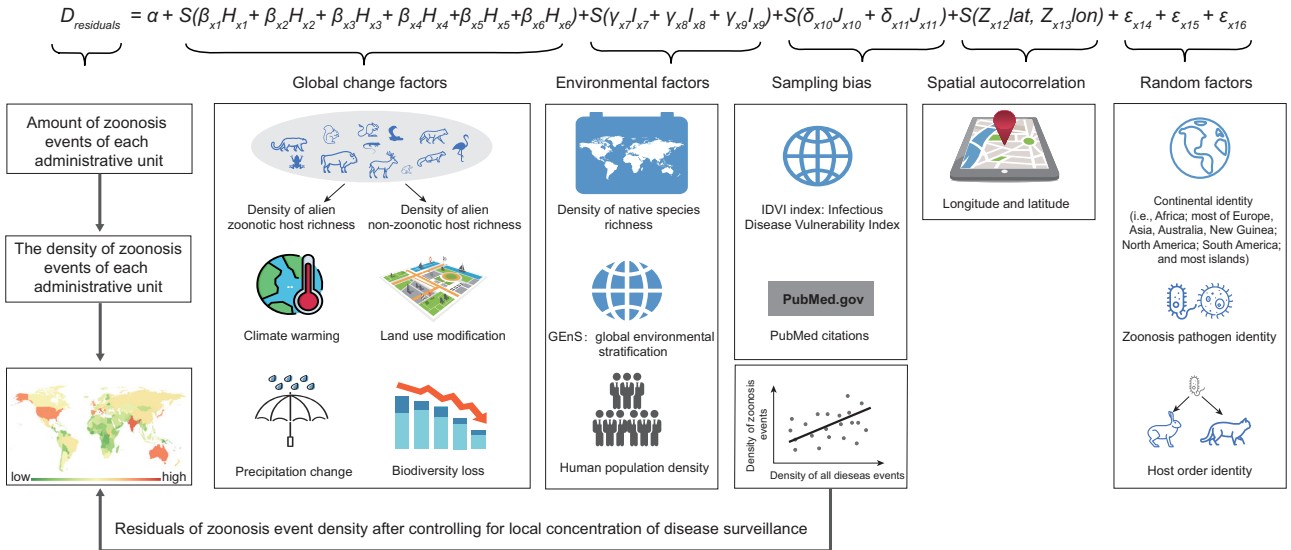

**Fig. 1 Analysis diagram to estimate the effect of alien animal invasions on zoonosis emergences at the global scale.** The role of alien animal zoonotic hosts was determined by accounting for global change factors ($H_{x1-x6}$), environmental factors ($I_{x7-x9}$), sampling bias ($J_{x10,11}$), spatial autocorrelation ($L_{lat, lon}$), and a lack of independence among zoonoses by treating pathogen, host order and continent as random factor intercepts ($\varepsilon_{x14-x16}$). Thin-plate spline smooths for each predictor variable are designated by $S()$, and $\alpha$, $\beta$, $\gamma$, $\delta$, and $Z$ are constants ($\alpha$ is an intercept and $\beta$, $\gamma$, $\delta$ and $Z$ represent the coefficient estimates of different predictor variables, and $\varepsilon$ represents the random effects). Silhouettes were freely obtained from "islide" plug-in (https://www.islide.cc).

Fig. 1). We applied Akaike Information Criterion (AIC) to select the most highly supported models (i.e., ΔAIC ≤ 2).

There were six variables that appeared in the top five most supported models: alien zoonotic host richness, human population density, biodiversity loss, temperature change, land-use change, and PubMed citations (Fig. 3A). The 95% confidence intervals (CIs) of the coefficients for these six predictors also did not overlap with zero in any model (Fig. 3B). In addition to alien host richness, human population size and PubMed citations explained the highest percentages of deviance in the number of zoonosis events, followed by the three global change factors (Fig. 3A). Most variables had nonlinear but generally positive relationships with the number of zoonosis events after accounting for other cofactors (Fig. 4). Although it was only included in three of the top five models, the latitude of a country was also a significant predictor of zoonosis events (Fig. 3A), with zoonosis events being more common in higher latitudinal areas (Supplementary Fig. 2). These results were independent of whether we used 10, 8, or 6 knots in the GAMM analyses (Supplementary Fig. 3).

Next, we added to the model an interaction between host order and alien zoonotic host richness to test whether certain orders were more important contributors to emergences of zoonoses. The effects of alien zoonotic host richness for three mammalian orders (i.e., Carnivora, Effect size ± S.E.: 0.0150 ± 0.0028, $P < 0.001$; Artiodactyla, 0.011 ± 0.0023, $P < 0.001$; and Rodentia, 0.0197 ± 0.0021, $P < 0.001$), three avian groups (i.e., waterfowl, 0.0149 ± 0.0022, $P < 0.001$; Galliformes, 0.0059 ± 0.0024, $P = 0.0134$; and Passeriformes, 0.0082 ± 0.0019, $P < 0.001$), and order Diptera of the invertebrates (0.0151 ± 0.0053, $P < 0.01$) had significantly stronger associations with zoonosis events than other host groups (Columbiformes, −0.0043 ± 0.0032, $P = 0.891$; Lagomorpha, 0.0033 ± 0.0045, $P = 0.462$; Psittaciformes, 0.0027 ± 0.0028, $P = 0.327$; amphibians, 0.0061 ± 0.0062, $P = 0.326$; reptiles, 0.0007 ± 0.006, $P = 0.902$, Fig. 5).

Finally, we plotted the estimated contribution of alien zoonotic host introductions to historical zoonosis events for each administrative unit globally predicted by the GAMMs with and without the alien zoonotic host (Fig. 6). This map suggests that zoonosis hotspots due to past alien zoonotic host introductions were most concentrated in Europe, Oceania, and the Caribbean islands (Fig. 6),

which is largely consistent with the global hotspots of alien animal establishment[26].

**Temporal relationship between alien animal invasions and zoonotic diseases**. To verify that there was a positive association between alien host introductions and zoonosis events through time, we first conducted a multiple regression analysis where we treated year as the replicate, the number of zoonotic diseases as the dependent variable, and the number of zoonotic and non-zoonotic alien introductions as the independent variables. Model averaging analyses of the generalized additive models showed that the number of zoonosis events through time was positively correlated with the number of alien zoonotic host introductions through time (Estimate = 3.04, 95% CI = 2.23~3.85, $P < 0.001$, Supplementary Fig. 4), but was not correlated with the number of alien non-zoonotic host introductions (Estimate = 0.62, 95% CI = −0.69~1.94, $P = 0.345$). In addition, we also conducted breakpoint regression analyses to evaluate whether the breakpoints (when there is a sharp change in a response variable in time) for zoonosis events and introductions of zoonotic and non-zoonotic alien hosts tended to coincide in time. The number of zoonosis events increased sharply in 1962 (i.e., a breakpoint based on AICc), just two years after alien zoonotic host introductions sharply increased in 1960 (Supplementary Fig. 5A, B). In contrast, the breakpoint for non-zoonotic alien hosts was in 1948 (Supplementary Fig. 5C), which was far from the zoonosis breakpoint in 1962.

**Discussion**

The present study, to the best of our knowledge, provided the first comprehensive global evaluation of the relationship between alien species invasions and zoonotic disease emergences. Our literature and database review showed a high diversity of zoonoses with an average of ~5.9 diseases per alien zoonotic hosts. This is likely an underestimate as our criteria for zoonotic alien hosts were highly conservative (Supplementary Data 2). Given taxonomic and geographic biases in zoonosis sampling, there might be many unknown zoonotic hosts that have not yet been reported or studied[27,28]. Despite this, we tried to account for potential sampling biases in disease surveillance efforts and which species and geographic locations were studied to

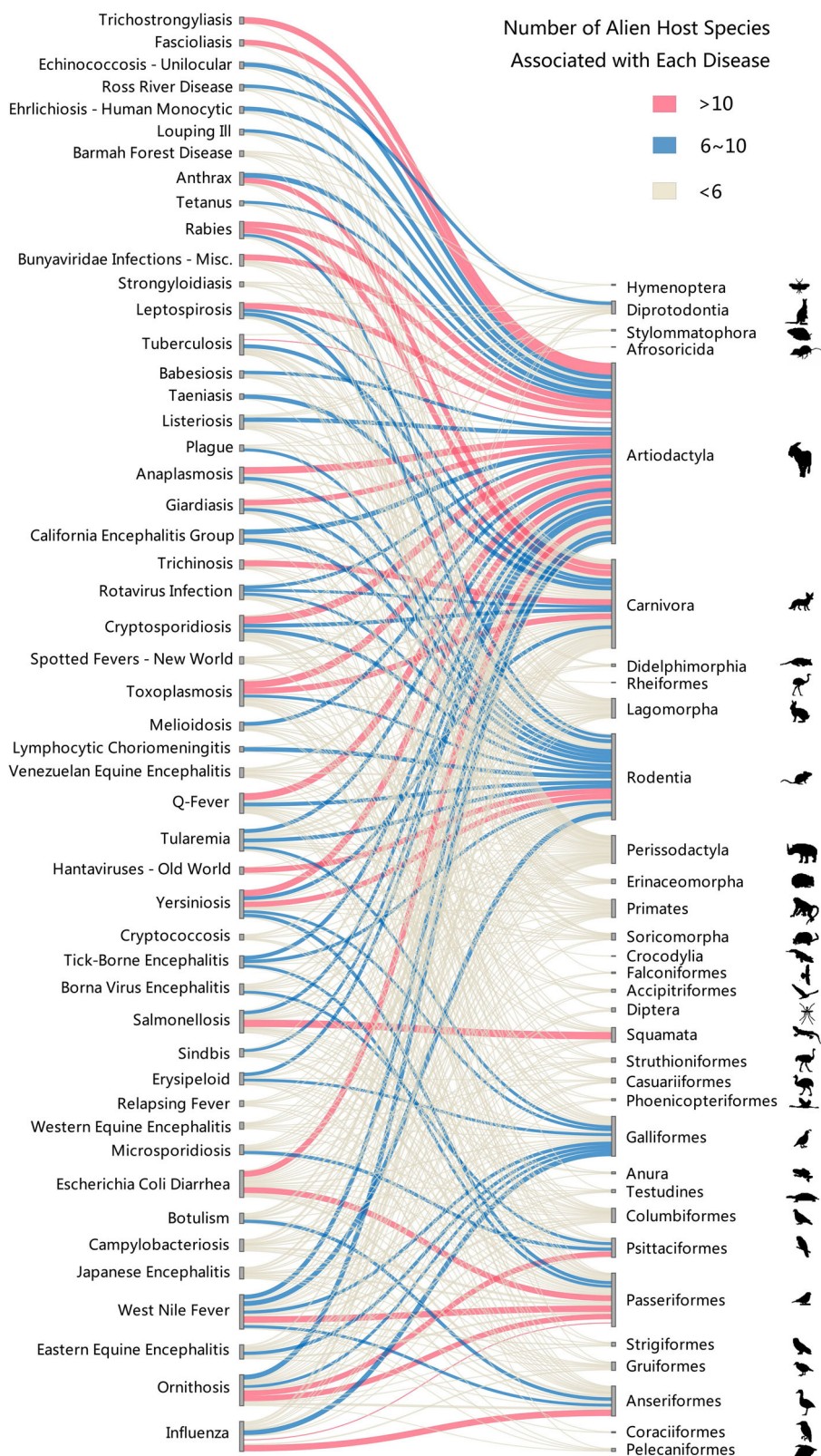

**Fig. 2 Associations between the zoonotic diseases reported by the GIDEON database and alien zoonotic hosts.** Bipartite network analysis shows the relatedness between the top 50 zoonoses with the largest number of alien zoonotic host species and the alien zoonotic host orders. The exact number of zoonotic diseases per alien zoonotic host is provided in Supplementary Data 3. Width indicates the number of zoonotic diseases carried by alien zoonotic host species in each order. The order of figure column is based on the default output of the R software to minimize the number of crossovers. Animal silhouettes were obtained from PhyloPic.

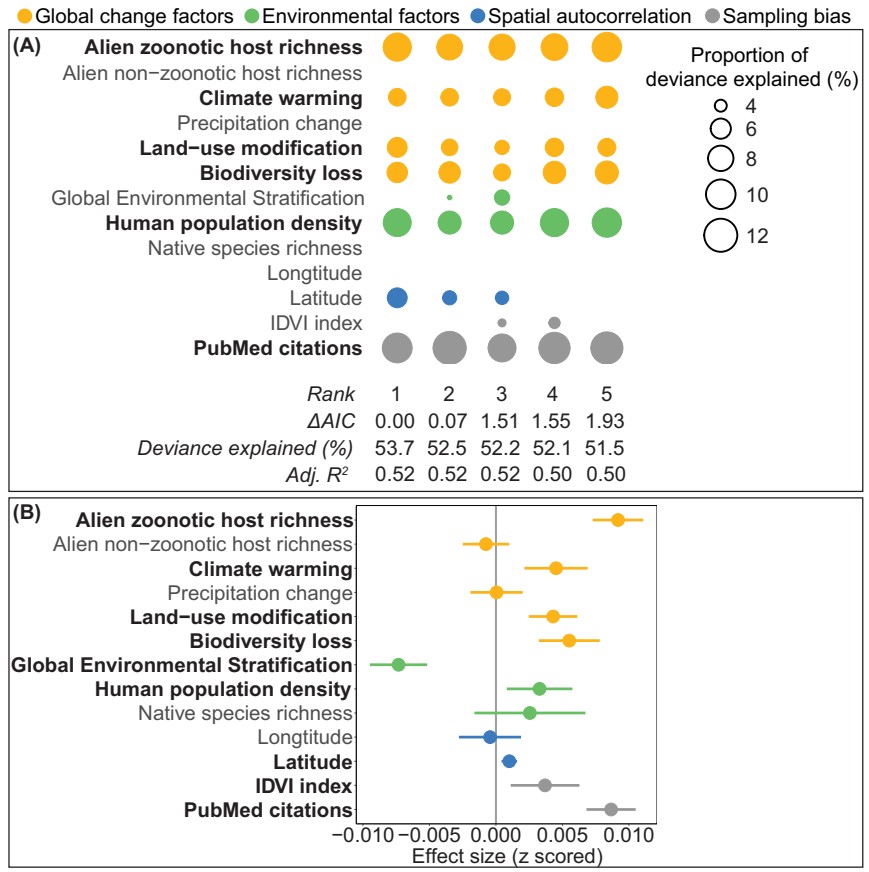

**Fig. 3 Proportion of deviance explained and effect size of each predictor variable in model averaging analyses based on GAMMs.** Columns represent individual models and rows represent predictor variables, with smoothing function knot value = 10. Shown in bold are the variables that appear in all five of the most highly supported models in panel **A** and that have model-averaged 95% confidence intervals that do not overlap zero in panel **B**. The circle size in panel **A** represents the proportion of deviance explained by each predictor and any blanks indicate that the predictor is not included in the model. The panel **B** represents mean effect sizes with 95% confidence intervals of different predictor variables explaining the number of zoonosis events worldwide (n = 10,473, Supplementary Data 4).

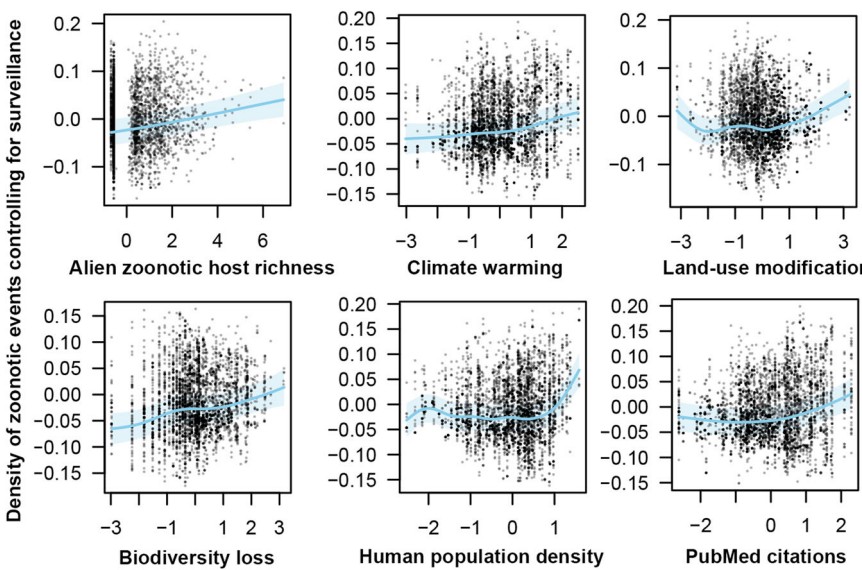

**Fig. 4 Relationships of the six most important predictor variables with zoonosis emergences in the five highly supported models.** Scatter plots represent the partial residuals of each smoothed variable when controlling for other variables. Blue lines show the predicted function of each variable with mean and the shaded area as the 95% confidence band based on GAMMs. The dependent variable (zoonosis event density) is treated as the residuals of the fitted regression correlating the density of zoonosis events and the density of all disease events to account for the degree of overall disease surveillance (Supplementary Data 4). All predictor variables were standardized (to a mean of zero and standard deviation of one) before entering the model.

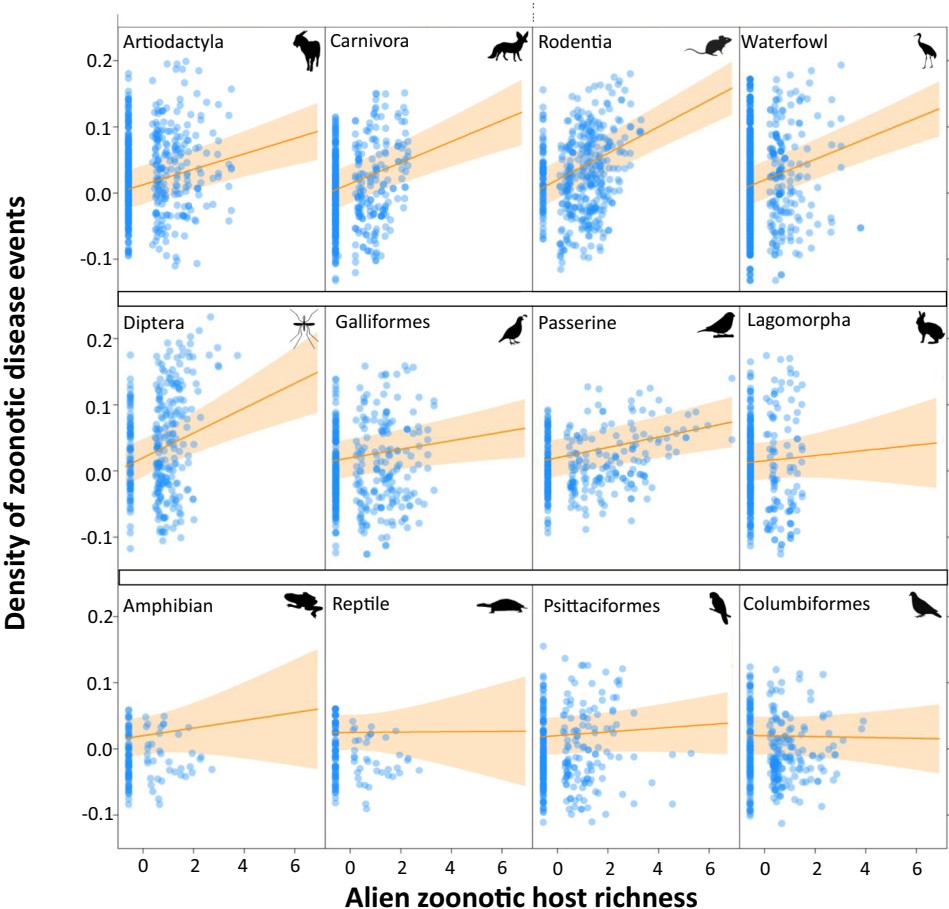

**Fig. 5 The relationship of alien zoonotic animal richness and the zoonosis emergences across alien host groups.** The strength of each host group is tested by including an interaction between the richness of alien zoonotic species with each of the 12 host orders identified as important in the GAMMs after accounting for other co-factors. Each host order had established alien populations in at least 50 administrative units. Lines show the predicted relationship between density of zoonotic disease events and alien zoonotic host richness, with mean and the shaded area as the 95% confidence band (Supplementary Data 4). All predictor variables were standardized (to a mean of zero and standard deviation of one) before entering the model. Animal silhouettes were obtained from PhyloPic.

increase the robustness of our findings. We encourage future studies to further support or refute our findings with more zoonoses and host discoveries.

Consistent with our hypothesis, we showed that the richness of alien zoonotic hosts was indeed an important factor determining the number of zoonosis events. Comparatively, we did not find an important effect of the richness of non-zoonotic host on the emergence of zoonoses (Fig. 3), demonstrating that the correlation of zoonosis events with alien zoonotic hosts was not simply a by-product of anthropogenic-related propagule pressure. We also detected a close temporal relationship between zoonosis emergences and alien zoonotic host introductions. These results corroborate the spatial patterns and further bolster the hypothesis that alien species invasions have contributed to the increase of zoonosis events over the last sixty years.

Our study illustrated that alien species in certain mammalian orders, such as Carnivora, Artiodactyla, and Rodentia, were significantly associated with zoonosis events. This result is consistent with previous studies showing that mammalian orders Carnivora, Artiodactyla, and Rodentia are strongly associated with zoonotic diseases[23,25]. Orders Chiroptera and Primates were also previously shown to be associated with human zoonotic diseases[23,25], but they were not included in our analyses because there are too few alien species from these orders. Many mammals have a predilection for human-dominated environments[29,30] and are closely related to

humans phylogenetically, both of which can facilitate spillover probabilities[23]. Within class Aves, there were significant associations between alien zoonotic host species introductions and zoonotic diseases for waterfowl (including five orders: Anseriformes, Gruiformes, Pelecaniformes, Phoenicopteriformes, Suliformes), and orders Galliformes and Passeriformes, but not Columbiformes and Psittaciformes (Fig. 5). Consistent with these findings, alien waterfowl can be carriers of cryptosporidiosis, giardiasis, and microsporidiosis[8], and a recent global study showed that passerine species were highly associated with human-disturbed habitats, which may increase the probability of passerine-related zoonosis events[30]. We found that the invertebrate zoonoses were dominated by Dipteran-hosted diseases. As examples, the rapid expansion of Aedes and Anopheles mosquitoes has resulted in the worldwide transmission of various diseases, such as malaria, yellow fever, dengue, chikungunya and lymphatic filariasis[31]. We did not detect effects of alien herpetofauna host species on zoonotic diseases consistent with there only being a few zoonoses shared by humans and herpetofauna[32] and the considerable physiological differences between herpetofauna and humans. Nevertheless, we suggest that the potential zoonotic risks from alien reptiles and amphibians should not be overlooked given continued alien reptile and amphibian introductions[33] and an increasing frequency of direct contacts between herpetofauna and humans through the pet trade and aquaculture[32].

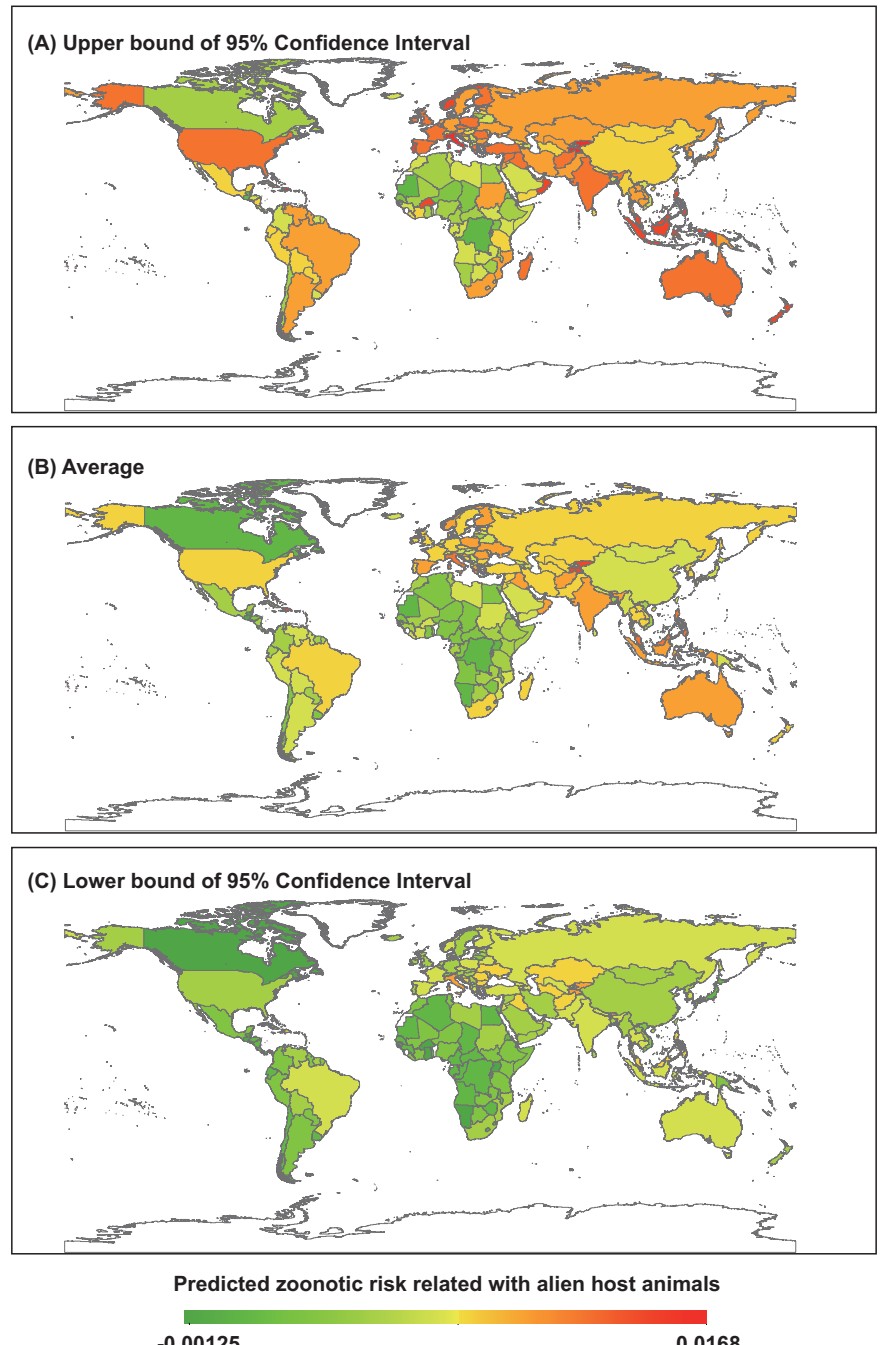

**Fig. 6 Global map showing the potential contribution of alien zoonotic host introductions to zoonosis events of each administrative area at the global scale.** Maps are derived for predicted zoonosis events caused by alien host species and the relative risk is calculated by subtracting the GAMM fitted values excluding the zoonotic host introduction from those using all predictor variables in Supplementary Data 4.

Our analyses also unsurprisingly unveiled that human population density was positively associated with zoonotic diseases. Countries with higher human population densities may be likely to host more zoonoses, which has long been observed by epidemiologists[1]. Consistent with past studies[21,34], we detected positive associations between the number of zoonosis events and both anthropogenic land-use transformation and changing temperatures. Anthropogenically influenced habitats can lead to more human-wildlife contacts that can facilitate spillover of zoonoses from wildlife to humans[30]. Similarly, insect vector-borne diseases can be favored in human-disturbed habitats where opportunities for human contact can increase[35]. Additionally, habitat modification can create more

vacant niches, facilitating establishment of zoonotic host populations[36]. Changing temperatures can expand the distributional ranges of pathogens, making previously unsuitable regions habitable[37], and can accelerate the development and increase the abundance, survival, and transmission rates of reservoir hosts or vectors[38]. We found that biodiversity loss rather than native host species richness was an important predictor of zoonosis events (Fig. 3), which was consistent with previous studies showing that there was no overall effect of biodiversity on zoonosis emergence at large spatial scales[39,40] and that anthropogenic biodiversity loss rather than natural biodiversity gradients was key in driving zoonoses[19,41]. The important role of PubMed citations in explaining

the number of zoonosis events in each administrative region confirmed that controlling for research effort or bias towards particular hosts and pathogens is important for understanding observed spatial patterns of disease[1,21,28].

Our results provide quantitative, global, spatial, temporal, and multi-taxonomic evidence supporting the hypothesis that established alien species can promote zoonosis events across the planet. The detected associations between zoonosis events and climate change, land-use change, biodiversity loss, and introduced species also offer a deeper understanding of the potential responses of zoonosis emergences to ongoing global change and might help inform policy and regulatory recommendations targeted at both the control of zoonotic diseases and the mitigation of the effects of global change. The provided global map identifying the likely contribution of introduced species to historical zoonosis events (Fig. 6) offers a decision support tool for directing both surveillance and disease control efforts. Assuming a positive correlation between past and future hubs of zoonotic diseases, the hotspots identified on this map should serve as a preliminary prediction of the locations where future diseases associated with zoonotic host introductions are most likely to occur and where zoonosis surveillance might be most wisely targeted.

## Methods

**Disease data source.** All analyses were conducted at the administrative level, and the exact list of known zoonotic diseases is recorded in the GIDEON database[22]. GIDEON is currently the most comprehensive and frequently updated infectious disease outbreak database reporting epidemics of human infectious diseases at the global scale and has been widely used in global zoonosis studies[42,43] (Last access date, November 9, 2020). The administrative designations used in our analyses were based on the Global Administrative Areas (GADM) database (www.gadm.org, downloaded on November 8, 2020), which includes very detailed boundary data for global countries and major island groups.

### Pattern and correlates of zoonosis events worldwide

*Number of zoonosis events.* GIDEON defines human infectious disease reservoirs as any animal, plant, or substrate supporting the survival and reproduction of infectious agents and promoting transmission to potential susceptible hosts. Its host category therefore includes all human-specific, zoonotic, multihost, and environmental agents. As our main aim was to test the role of established alien animal species in the emergence of zoonotic diseases, we focused on a total of 161 diseases specified in GIDEON's host designations and definitions as nonhuman zoonotic ($n = 115$) and multihost ($n = 46$) diseases (Supplementary Data 1) and excluded diseases with human-specific hosts that do not need animals to persist or be transmitted. The infectious agents of nonhuman zoonotic diseases complete their entire lifecycle in nonhuman hosts but may have the potential to spillover and infect human populations. Infectious agents of multihost diseases can use both human and animal hosts for their development and reproduction. We measured the number of zoonosis events for each jurisdiction according to five host taxonomic groups: mammals, birds, invertebrates, reptiles and amphibians. These zoonoses were mainly caused by bacteria, viruses, parasitic animals and fungi. We excluded zoonoses from the Algae (3 diseases) due to low sample sizes in GIDEON.

### Correlates of the number of zoonosis events

*Climatic variables.* Following a previous study[21], we used global environmental stratification (GEnS) as a composite bioclimatic variable generated by stratifying the Earth's surface into zones with similar climates[44]. The GEnS database was constructed based on a total of 125 strata across 18 global environmental zones with a spatial resolution of 30 arc seconds (equivalent to approximately 0.86 km² at the equator). The values in GEnS range from 1 to 18 with a higher value indicating warmer and wetter conditions.

*Human population density.* We used human population density as one general anthropogenic factor reflecting propagule pressure and human-assisted pathogen movements[1,21,45]. Human population size data and the land area of each jurisdiction were collected from World Bank Open Data from 2011 to 2020 (available at https://data.worldbank.org/indicator/SP.POP.TOTL, accessed on November 18, 2020). We then calculated the human population density using the human population size divided by the land area.

*Native potential host richness and biodiversity loss.* Data on the richness of native amphibians, birds, and mammals were derived from the Biodiversity Mapping website (https://biodiversitymapping.org/wordpress/index.php/home/, accessed on August 19, 2020), which were based on studies from Jenkins et al. (2013)'s and Pimm et al. (2014)[46,47]. The map of reptile diversity is based on an updated database

of the global spatial distribution of reptiles[48]. All diversity maps for each taxon were generated through the calculation of grid-based richness at a spatial resolution of 10 km × 10 km in ArcGIS[46]. We did not include native invertebrate richness, as global maps for most invertebrate taxa are not yet available. For the loss of native biodiversity, we followed the previous study by first extracting the list of threatened species (NT, EN and VU categories evaluated by the IUCN Red List, access on May 10th, 2021)[29], and then calculated the number of threatened species for each taxon distributes in each administrative unite as a proxy of biodiversity loss.

*Richness of established alien zoonotic host species.* We quantified the richness of established alien animal species from the five main taxonomic groups (mammals, birds, reptiles, amphibians and invertebrates) based on 4,522 establishment events of 795 alien animals in each of 201 jurisdictions according to various databases. Data on 262 established alien reptiles and amphibians were compiled from multiple publications, including Kraus's compendium[49] and other recent updates[50]. Data on 337 established alien birds after removing all migratory bird species as vagrants were collected from the Global Avian Invasions Atlas (GAVIA)[51], which is a comprehensive database of the global distribution of established alien birds. Data on 119 established alien mammals were obtained from the Introduced Mammals of the World database[52] and the more recent update[53]. Data on 77 terrestrial alien invertebrates (66 insects and 11 other groups) across 7 taxa with native and invaded range information were obtained from the Global Invasive Species Database (GISD, http://www.iucngisd.org/gisd/, accessed on July 1, 2020). We calculated the richness of both zoonotic and non-zoonotic alien host species for each order. We first conducted an intensive literature review for each established alien species of each of the four taxa to determine whether they transmit pathogens to humans (Supplementary Data 2). The identification of zoonotic or non-zoonotic host may be influenced by under-sampling in the literature. We therefore incorporated the latest synthesis of human-infecting pathogens in the 'CLOVER' dataset to identify zoonotic and non-zoonotic animal hosts[54]. The CLOVER dataset compiled GMPD2[55], EID2[56], HP3[23] and Shaw[57] databases and is currently the most comprehensive dataset on host-pathogen associations. Based on this information, we then categorized each alien species as a 'zoonotic host' or 'non-zoonotic host'. The records of the established alien species were assigned to GADM jurisdictions, and we calculated the richness of the established alien zoonotic and non-zoonotic host species for each taxonomic group within each jurisdiction. In order to increase the statistical power, we conducted subsequent modeling analyses based on four mammalian orders (i.e., Carnivora, Cetartiodactyla, Lagomorpha, and Rodentia), five avian groups (i.e., waterfowl including five orders: Anseriformes, Gruiformes, Pelecaniformes, Phoenicopteriformes and Suliformes; Columbiformes, Galliformes, Passeriformes, Psittaciformes), the order Diptera of the invertebrates, and herpetofauna as a whole, which have established alien populations in at least 50 administrative units.

*Climate change.* We extracted historical monthly mean temperature and precipitation data recorded between 1901 and 2009 from the University of East Anglia Climate Research Unit (CRU, https://sites.uea.ac.uk/cru/, accessed on November 30, 2020)[58]. This database provides historical global-scale yearly climatic data with the finest resolution of 0.5° grids. We generated the temperature and precipitation values for all grids in each jurisdiction, calculated the slope of the temperature and precipitation for the time series of the years 1901 to 2009 for each grid and generated the averages based on all grids within each jurisdiction.

*Anthropogenic land-use change.* We downloaded global land-use data from the Anthromes v2 Dataset (Anthropogenic Biomes version 2, accessed on October 15, 2020) in ESRI GRID format[59]. We used the 1900 and 2000 data to calculate the temporal changes in land use. By using the *reclassify* and *raster* function in ArcGIS, we calculated the percentage of grids in which the land-use type changed to a more anthropogenically influenced type from 1900 to 2000 for each jurisdiction, including 15 scenarios: Wildlands to Seminatural, Wildlands to Rangelands, Wildlands to Croplands, Wildlands to Villages, Wildlands to Dense Settlements, Seminatural to Rangelands, Seminatural to Croplands, Seminatural to Villages, Seminatural to Dense Settlements, Rangelands to Croplands, Rangelands to Villages, Rangelands to Dense Settlements, Croplands to Villages, Croplands to Dense Settlements, and Villages to Dense Settlements.

*Sampling effort, reporting bias and incomplete data.* A potential issue in quantifying the effects of different predictor variables on the number of zoonosis events is the need to account for the differences in survey effort, reporting bias and incomplete disease data among regions[1,21,28]. There is a high probability that zoonosis discovery is spatially biased by uneven levels of surveillance across countries, as the global allocation of scientific resources has been focused on rich and developed countries. We thus included the Infectious Disease Vulnerability Index (IDVI), which is a comprehensive metric reflecting the demographic, health care, public health, socioeconomic, and political factors that may have an impact on the capacity of surveillance and detection of infectious diseases in each country[60]. Second, we followed the methods of a previous study[21] to control for reporting biases. We incorporated PubMed citations per disease for each jurisdiction using a Python-based PubCrawler[21]. In addition, we added the longitude and latitude of the geographic centroid of administrative units to control for spatial

autocorrelation as there would be a higher probability of having similar diseases in nearby than distant administrative units[61].

**Statistical analysis**. The number of zoonosis events, native potential host richness, established alien animal richness and human population density were log-transformed to improve linearity. A potential issue in our data analysis is that the numbers of zoonosis events and the numbers of native and alien animal species are strongly influenced by geographical area, as larger countries or regions may host more native or alien animal species and more disease events. We therefore calculated the density of native or alien species richness and the number of zoonosis events using the total number of zoonosis events divided by the geographical area of each jurisdiction. Furthermore, the number of zoonosis events may also be influenced by the degree of local disease surveillance. We thus obtained the residuals from a regression correlating zoonosis event density and all disease event density, and used them as the dependent variable for further analyses (Fig. 1). As some of our variables may be expected to be nonlinear, we performed generalized additive mixed model (GAMM) analyses following Mollentze & Streicker 2020's framework[25] to quantify the relationships between different predictor variables and the number of zoonosis events. We started with a full model with zoonosis event density controlling for overall disease surveillance as the response variable and 13 smoothed fixed effects (Fig. 1 and Supplementary Data 4): GEnS, human population density, density of native species richness, biodiversity loss, density of alien zoonotic host richness, density of alien non-zoonotic host richness, climate (temperature and precipitation) change, land-use change, IDVI, PubMed citations, longitude and latitude of geographic centroid of administrative units. The reason why we included the density of alien non-zoonotic host richness as a covariate is because this variable can serve as a positive control for propagule pressure, allowing us to more explicitly test whether zoonotic alien hosts contribute to zoonoses beyond propagule pressure associated with non-zoonotic alien hosts, which cannot directly increase zoonotic diseases. These predictor variables were not highly collinear as their correlation coefficients based on Pearson rank correlation analyses were all <0.65 (Supplementary Fig. 6). Because human history may have a great influence on disease outbreaks, as there may be more human pathogens on continents subject to earlier human settlement[45], we followed this literature by including continental identity (i.e., Africa, region of origin and first settlements; most of Europe, Asia, Australia, New Guinea, by approx. 40,000–60,000 BP; North America, by approx. 15,000–25,000 BP; South America, by approx. 1000–5000 years after North America, i.e., 10,000–24,000 BP; and most islands, by approx. 1000–7000 BP) as a random intercept to control for potential pseudoreplication. In addition, to account for the lack of complete independence among disease events caused by the same pathogens or associated with the same introduced host orders (Fig.1), we included pathogen identity and host order as two additional random intercepts. We fitted all models using restricted maximum likelihood method and ranked all candidate models by the Akaike's Information Criterion (AIC) theoretic approach[25,62]. Models including all possible combinations of the 13 predictor variables (total $2^{13}$-1 = 8191 models) were ranked, and the models within 2 AIC unites (i.e., $\Delta$AIC ≤ 2) compared with the top model were considered to be highly supported[62]. For each model, we computed the standardized estimates of the regression coefficients of the predictor variables with the 95% confidence intervals (CIs), and considered effects statistically significant when the 95% CIs did not overlap zero. In addition to variable significance, we also calculated the proportion of the deviance explained by each predictor variable by comparing the sub-models in the absence of the variable to the full model[25]. To better compare the coefficients of the different covariates, we standardized each of the predictor variables to a mean of zero and standard deviation of one before it was entered into the model[63]. Furthermore, we used different levels of thin-plate smoothers with 6, 8, and 10 knots for the fixed-effect variables in GAMMs[25]. As the results were similar regardless of which knot we used, we present the results from the analyses with 10 knots in the main text but provide the results with 6 and 8 knots in the supporting materials (Supplementary Fig. 3). All analyses were conducted in the *gamm4, mgcv, visreg, dplyr,* and *MuMIn* packages in R version 4.0.3[64] (Supplementary Notes).

To further test whether there are different responses of the zoonosis emergences among alien host groups, we fit an interaction between host order and alien species richness to investigate whether the effect of alien zoonotic host richness varied across taxonomic groups. Finally, to evaluate the potential contribution of alien zoonotic host introductions on historical zoonosis events for each administrative unit, we generated the fitted values along with 95% CIs of the number of zoonosis events in each administrative unit predicted by using the predictors in GAMMs, and subtracted the predicted values excluding zoonotic host introductions from those using all predictor variables.

The observed spatial correlation of zoonosis emergences with alien animal invasions might be problematic because there may be mismatch in the occurrence of alien animal invasions and zoonosis events in time. For instance, zoonosis events at a location might have occurred earlier than the alien animal invasions, which could not have caused the disease despite a strong spatial correlation. We therefore further explored the temporal relationship of alien zoonotic (and non-zoonotic) host introductions and zoonotic diseases over years. To achieve this, we collected the introduction time of each alien zoonotic and non-zoonotic host species in each country or region based on the alien avian introduction database and literatures from alien birds[51,63], the Introduced Mammals of the World database[52] for alien mammals, and the Global Invasive Species Database (GISD) for alien invertebrates. We obtained the number of new zoonosis events over time from the GIDEON database (Last access date, November 9, 2020), and calculated the number of zoonosis events in each year for different taxa. For each year,

the number of new zoonosis events divided by the number of alien host introductions for each country or region was used to evaluate the magnitude of the relationship between alien zoonotic host introductions and the number of new zoonosis events. We then used two approaches to analyze their temporal relationships. Firstly, we conducted a generalized additive modeling analysis where we treated year as the replicate, and zoonotic and non-zoonotic alien introductions as the independent variables, and the number of zoonosis events as the dependent variable, to explore the general trend of alien animal host introductions and the number of zoonosis events along time. Additionally, we applied breakpoint regression analyses using the *segmented* package in R (Supplementary Notes). In these analyses, the identified the breakpoint reflects the year in which there was a rapid increase in the number of zoonotic or non-zoonotic host species or the number of zoonosis events. For these analyses, we fit the left-horizontal regression and two-slope regression that are widely used in ecological and biogeographical studies[65], and applied an AIC-based approach to identify the optimal breakpoint year. We combined data across various host taxa for the temporal analyses because some orders had little data on the timing of alien species establishment.

**Reporting summary**. Further information on research design is available in the Nature Research Reporting Summary linked to this article.

## Data availability
The data used in this study are from existing datasets and are included in Supplementary Data 1-4. The exact list of known zoonotic pathogens and zoonoses at the administrative level is recorded in the GIDEON database (www.gideononline.com) (Last access date, November 9, 2020). The administrative designations used in our analyses are based on the Global Administrative Areas (GADM) database (www.gadm.org, downloaded on November 8, 2020). The alien host-pathogen association data are based on Gibb's dataset (CLOVER_Associations_Initial.csv, https://doi.org/10.5281/zenodo.4435128) and other literatures in Supplementary Data 2. The Infectious Disease Vulnerability Index (IDVI) is from Moore et al. (2017) (https://pubmed.ncbi.nlm.nih.gov/28845357/). We use global environmental stratification (GEnS, https://www.geoportal.org/) as a composite bioclimatic variable generated by stratifying the Earth's surface into zones with similar climates. For the temperature and precipitation change variable, we extract historical monthly mean temperature and precipitation data recorded between 1901 and 2009 from the University of East Anglia Climate Research Unit (CRU, https://catalogue.ceda.ac.uk/uuid/3f8944800cc48e1cbc29a5ee12d8542d, accessed on November 30, 2020), and global land-use data from the Anthromes v2 Dataset (Anthropogenic Biomes version 2, https://ecotope.org/anthromes/v2/, accessed on October 15, 2020) in ESRI GRID format. Human population size data and the land area of each jurisdiction are collected from World Bank Open Data from 2011 to 2020 (available at https://data.worldbank.org/indicator/SP.POP.TOTL, accessed on November 18, 2020). Data on the richness of native amphibians, birds, and mammals are derived from the Biodiversity Mapping website (https://biodiversitymapping.org/wordpress/index.php/home/, accessed on August 19, 2020) derived from Jenkins et al. (2013) (https://doi.org/10.1073/pnas.1302251110) and Pimm et al. (2014) (https://www.science.org/doi/10.1126/science.1246752). The latest map for the reptile species is from Roll et al. (2017) (https://doi.org/10.5061/dryad.83s7k). The list of threatened species evaluated as NT, EN and VU used for calculating the loss of native biodiversity is extracted from the IUCN Red List (http://www.iucnredlist.org, access on May 10th, 2021). The established alien species list and the introduction time information used for the temporal analyses are from Kraus's (2009) compendium (https://link.springer.com/chapter/10.1007/978-1-4020-8946-6_6) and Capinha et al. (2017) (https://doi.org/10.1111/ddi.12617) for reptiles and amphibians, the Global Avian Invasions Atlas (GAVIA) (https://doi.org/10.6084/m9.figshare.4234850) for established alien birds, Long's (2003) book (https://ebooks.publish.csiro.au/content/introduced-mammals-world) and Capellini et al. (2015) (https://doi.org/10.1111/ele.12493) for established alien mammals, and the Global Invasive Species Database (GISD, http://www.iucngisd.org/gisd/, accessed on July 1, 2020) for established alien invertebrates.

## Code availability
The R code for running the analyses is provided in Supplementary Information file (Supplementary Notes).

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

## Acknowledgements
This work is supported by the National Natural Science Foundation of China (32171657, 31870507) to X.L., the Third Xinjiang Scientific Expedition Program (Grant No. 2021xjkk0600) to Y.D., the Youth Innovation Promotion Association of Chinese Academy of Sciences (Y201920), and the Grant of High Quality Economic and Social Development in South Xinjiang (NFS2101) to X.L.

## Author contributions
X.L. is the lead contact for this paper. X.L. conceived the project. X.L. and J.R.R. designed the study. X.L. and Q.H. supervised the project. L.Z., R.C., Y.X., X.L., L.H., J.L., X.W., X.Y., and Z.W. collected the data. X.L., L.Z., S.G., and Y.D. performed the data analyses. X.L., J.R.R. and L.Z. wrote the manuscript.

## Competing interests
The authors declare no competing interests.
