## [Peer Review File · Nature Communications]

Reviewers' Comments:

Reviewer #1:

Remarks to the Author:

This is an interesting paper addressing an important issue – the extent to which human zoonotic emerging infectious diseases are associated with alien zoonotic host introductions on a global scale. Overall, I found it to be clearly written and very well presented. The overall conclusion is interesting – that zoonotic EID events are associated with the richness of alien mammalian species (especially amongst carnivores, rodents and artiodactyls) amongst birds with waterfowl, Galliformes and passerine birds and Diptera.

However, a major issue that concerns me is that the response variable is 10,472 zoonotic EID events (line 75), but these are derived from only 161 distinct human pathogens (line 101). Although I can't see information presented in the main text or in the supplementary materials, my strong suspicion is that the EID events arising from rodents are dominated by a single species (*Rattus rattus*), and that those arising from Diptera are dominated by mosquitoes (and within the mosquitoes, I expect *Aedes aegypti* plays a particularly important role). I expect for the other host taxonomic groups, a relatively small number of species likewise will dominate the data. It would be quite simple to include some information on the identity of the major pathogens considered and the major zoonotic hosts, either in the main text or in the supplementary materials and I think this should be done. My suspicion is that it may not be the richness of the alien fauna driving the relationship, but the presence of a small number of key species.

There is obviously the potential for very substantial nonindependence within the dataset, both in terms of the pathogens and the zoonotic hosts).

This may be perhaps have been dealt with in the analysis – line 358 states that GLMM was used, which would be an appropriate way to handle the problem. However, the statistical analysis section does not contain an adequate description of how the analysis was performed, particularly in terms of what the fixed and random effects were in the models. As far as I can see, only one random effect is explicitly identified (continental identity, see lines 368-373). I can't find any further and more detailed description of the statistical methodology in the supplementary materials

Reviewer #2:

Remarks to the Author:

Zhang et al present an interesting study linking biological invasions with the downstream threat of zoonotic diseases. Their analysis is intriguing and the models involved have been well-constructed and well-reported, and the study is very well-written. I think the study is excellently formulated and well worth publishing, and I believe the premise and findings, but I have suggestions to improve the models' inference, mostly to do with elements of sampling bias and data completeness.

Major comments:

- Although I respect the use of the IDVI and Lancet infectious diseases and JID citations to control for sampling biases, the authors need to take a look at their models and make sure that they actively account for both species-level sampling biases and local concentration of disease surveillance. A good way to do the former might be to incorporate the pubmed citation numbers of the species involved in each country's total (per e.g. Olival et al1 and many others). A way to do the latter might be to derive a measure of zoonotic EID outbreak concentration *in contrast to non-zoonotic outbreaks* to account for reporting heterogeneity. For example, fit a linear model correlating Zoonotic EIDs ~ all EIDs, and take the residuals from that to analyse them. That would go a long way to supporting their hypothesis separate from sampling bias. Alternatively, just fit overall EIDs into the model as an explanatory variable?
- Regardless, sampling bias should be more prominently discussed in the paper itself rather than just put in the methods, as I spent the time reading the paper wondering if everything was robust to these biases or not.
- The analyses could control more effectively for geographic non-independence in their results,

rather than merely using country-level metrics. Spatially heterogeneous sampling effort is important in determining observed disease patterns, and some recent analyses have included some very advanced approaches to deal with it. See for example the pubcrawler approach used by Allen et al² and the spatial autocorrelation analyses used by Albery et al³. Fitting smoothers or autocorrelation effects should be attempted, e.g. using the countries' centroids. Additionally, adding country size as a fixed effect might be informative.

- Similarly, is controlling for non-(observed) zoonotic host diversity really sufficient? Given taxonomic and geographic biases in sampling and the fact that the virome is woefully undersampled, is it possible that many of these "non-zoonotic species" just haven't been observed with zoonoses yet? Even if this measure is used, I appreciate the authors' use of a literature search to identify zoonotic and non-zoonotic hosts, but there are large databases that can be used to identify these species-level data easily. Most recently, Gibb et al synthesised and published a large meta-dataset that can be used for just this purpose⁴. I incorporated the authors' zoonotic list from Supplementary Table 2 into Gibb et al.'s data and found an extra >2000 animal hosts of human-infecting parasites that were not mentioned in their original list. I used the definition "were ever seen hosting parasites that have ever been found in humans", which is a fairly loose definition, but could provide a helpful contrast to their list of 800 or so species. I suggest incorporating this dataset, or if not, incorporating other open datasets of known host-parasite associations.
- While I appreciate the authors' desire to do analyses separately by order, I would recommend lumping the orders together, fitting the same variables, and then fitting an interaction effect between order and the effects of interest. This exercise will inform whether orders actually show *different* responses to the explanatory variables of interest, or whether it is parsimonious to keep them together. If the inclusion of the order interaction does not help (which I suspect might be the case) this will substantially simplify their reporting, which will help to include the analytical modifications I've outlined above.
- More generally, these are relatively simplistic generalised linear models, and there are doubtless non-linear relationships in there, so GAMMs would be preferable.
- Showing plots of the relationships between (at least) invasion numbers and EID numbers, including the fit (and 95% intervals) from the models would help to assess the validity of a linear fit, rather than just displaying the forest plot, similar to what they've done with Figure 3. This would also inform other aspects of the data distribution and would help to interpret the strength of the relationship and the variation accounted for by the effects. (given that I recommend removing the temporal analysis below, I would replace Figure 3 with the figure I'm suggesting here). Moreover, why have the authors fitted a smoother to Figure 3 if it's a relationship that they tested with linear fits in the main analyses? Why not fit the line from the model?
- P values, effect sizes, R², etc. should all be reported more visibly for all models, as should the model formulations in the methods. Having an explicit formula always helps. Most pressingly, although the invasion effect is significant, what proportion of the variance does it account for? These data could be combined with my suggested scatter plots including the model fits to increase their accessibility.

Line comments:

- 36: Can you add mentions of the taxa for which these effects were tested but not observed? Given the removal of important groups, it's unclear how valid these statements are and whether they really belong in the abstract.
- 54: needs comma after "aquaculture" or no comma before "such"
- 63-66: I think this passage could do with citing the Park lab's recent work on biological invasions and the factors driving parasite acquisition⁵ and Carlson et al.'s prediction that climate change-associated range shifts will drive cross-species transmission (and therefore zoonotic risk)⁶.
- 93: "theses" should be "these"
- 98: "both of" should be "both"
- 99: Again a good place to cite 5 rather than a review paper.
- 104-106: This order of continents is extremely what would be expected according to geographically heterogeneous sampling bias.
- 107: As is the finding that more occurred in higher latitudes.
- 133: This Keesing and Ostfeld study is a perspective, not a research article, and merely reports the orthodoxy. An Order-level observation like this should be attributed to a research article like the Olival paper or should be reconsidered according to the more thorough analyses by Nardus

Mollentze and Daniel Streicker⁷.

- 135: The fact that these important orders were removed from the analysis casts a bit of doubt on the validity of the findings reported in the abstract, as I mentioned above. How many orders were actually tested versus how many were found to be important? Choosing a correct null (per Mollentze and Streicker⁷) is very important for making Order-level statements like this.
- 173: "class" should be "classes"
- 190: Showing that two time series are correlated is actually extremely fraught statistically: <https://twitter.com/sethfinnegan1/status/1427358978766360577>. This analysis is not informative, and should be removed unless the authors can conduct a more believably unbiased analysis. If the spatial replicate approach can be maintained, that would work better.
- 294-297: See above major comment about this table.

1. Olival, K. J. et al. Host and viral traits predict zoonotic spillover from mammals. *Nature* 546, 646–650 (2017).
2. Allen, T. et al. Global hotspots and correlates of emerging zoonotic diseases. *Nat. Commun.* 8, 1124 (2017).
3. Albery, G. F., Carlson, C. J., Cohen, L. E. & Eskew, E. A. Urban-adapted mammal species have more known pathogens. (2021).
4. Gibb, R. et al. Data proliferation, reconciliation, and synthesis in viral ecology. *Bioscience* (2021) doi:10.1101/2021.01.14.426572.
5. Schatz, A. M. & Park, A. W. Host and parasite traits predict cross-species parasite acquisition by introduced mammals. *Proc. R. Soc. B* 288, (2021).
6. Carlson, C. J. et al. Climate change will drive novel cross-species viral transmission. *bioRxiv* (2020) doi:10.1101/2020.01.24.918755.
7. Mollentze, N. & Streicker, D. G. Viral zoonotic risk is homogenous among taxonomic orders of mammalian and avian reservoir hosts. *Proc. Natl. Acad. Sci.* 1–8 (2020) doi:10.1073/pnas.1919176117.

Point-by-Point Response to Reviewers' Comments:

Reviewer 1

Review Comments 1)

“This is an interesting paper addressing an important issue – the extent to which human zoonotic emerging infectious diseases are associated with alien zoonotic host introductions on a global scale. Overall, I found it to be clearly written and very well presented. The overall conclusion is interesting – that zoonotic EID events are associated with the richness of alien mammalian species (especially amongst carnivores, rodents and artiodactyls) amongst birds with waterfowl, Galliformes and passerine birds and Diptera.”

Our Response: Thank you very much for your positive comments and emphasizing the importance of our work.

Review Comments 2)

*“However, a major issue that concerns me is that the response variable is 10,472 zoonotic EID events (line 75), but these are derived from only 161 distinct human pathogens (line 101). Although I can't see information presented in the main text or in the supplementary materials, my strong suspicion is that the EID events arising from rodents are dominated by a single species (*Rattus rattus*), and that those arising from Diptera are dominated by mosquitoes (and within the mosquitoes, I expect *Aedes aegypti* plays a particularly important role). I expect for the other host taxonomic groups, a relatively small number of species likewise will dominate the data. It would be quite simple to include some information on the identity of the major pathogens considered and the major zoonotic hosts, either in the main text or in the supplementary materials and I think this should be done. My suspicion is that it may not be the richness of the alien fauna driving the relationship, but the presence of a small number of key species.”*

Our Response: The reviewer makes an excellent point here, and we are grateful for this input on our manuscript. We apologized that we did not provide the exact number of alien host species for each zoonosis, which may have confused the reviewer. We have carefully re-checked the 93,544 pathogen-host records gathered from the primary literature, databases and a newly added database recommended by the reviewer (Gibb et al. 2021) and have

verified the associations between the 161 zoonoses with the 795 established alien animal species. As you will see, although there are indeed some alien taxa (e.g., Artiodactyla, Rodentia, Carnivora, Passerine, Galliformes, waterfowl, and Dipteran) that can carry more zoonoses than others, overall we found that there was an average of about 13 alien zoonotic host species per disease (Line 118-129). We have provided the number of alien zoonotic host species for each zoonosis reported by the GIDEON database in Supporting Table 2.

Furthermore, to improve the visualization of this important information, we have added a new Sankey figure (new Fig. 1, Line 731) showing the quantitative relationship between the zoonoses and alien zoonotic hosts. We thank the reviewer for this great suggestion, which has made our findings clearer to readers.

Review Comments 3)

“There is obviously the potential for very substantial nonindependence within the dataset, both in terms of the pathogens and the zoonotic hosts). This may be perhaps have been dealt with in the analysis – line 358 states that GLMM was used, which would be an appropriate way to handle the problem. However, the statistical analysis section does not contain an adequate description of how the analysis was performed, particularly in terms of what the fixed and random effects were in the models. As far as I can see, only one random effect is explicitly identified (continental identity, see lines 368-373). I can’t find any further and more detailed description of the statistical methodology in the supplementary materials.

Our Response: The reviewer points out a very important issue on the potential nonindependence of our dataset. We are sorry that we did not explain this important issue clearly. In our revised manuscript, we have provided a new analysis diagram to show the whole analysis process, including the model formula, the dependent variable, and the fixed and random effects (new Fig. 2, Line 739). As you will see, we have included the host identity, the pathogen identity and the continental identity as random factors to prevent pseudo-replications in our new round of data analyses. We have provided these methodological details in our revised methods (Line 453-457) and updated all our results based on these new analyses (Line 138-171). Importantly, we obtained similar results on the importance of alien zoonotic host richness driving the number of zoonosis events.

Reviewer 2

General comments *“Zhang et al present an interesting study linking biological invasions with the downstream threat of zoonotic diseases. Their analysis is intriguing and the models involved have been well-constructed and well-reported, and the study is very well-written. I think the study is excellently formulated and well worth publishing, and I believe the premise and findings, but I have suggestions to improve the models’ inference, mostly to do with elements of sampling bias and data completeness.”*

Our Response: Thank you very much for your generally positive and constructive comments that have improved the quality of our work. We have revised our manuscript following each of your suggestions, especially the main concern regarding sampling bias and data completeness. Please see our point-to-point responses below.

Review Comments 1) *“Although I respect the use of the IDVI and Lancet infectious diseases and JID citations to control for sampling biases, the authors need to take a look at their models and make sure that they actively account for both species-level sampling biases and local concentration of disease surveillance. A good way to do the former might be to incorporate the pubmed citation numbers of the species involved in each country’s total (per e.g. Olival et al1 and many others). A way to do the latter might be to derive a measure of zoonotic EID outbreak concentration *in contrast to non-zoonotic outbreaks* to account for reporting heterogeneity. For example, fit a linear model correlating Zoonotic EIDs ~ all EIDs, and take the residuals from that to analyse them. That would go a long way to supporting their hypothesis separate from sampling bias. Alternatively, just fit overall EIDs into the model as an explanatory variable?”*

Our Response: The reviewer makes an excellent point here, and we are grateful for this input on our manuscript. In our revised manuscript, as suggested by your 3rd comment below, we have followed the methods of Olival et al. (2017) and controlled for species-level sampling biases by using a PubCrawler method to quantify the research effort of each country for each host species by searching PubMed for the number of disease-related publications. For the potential biases caused by the degree of disease surveillance, we followed the reviewer’s suggestion by obtaining the residuals from a model correlating zoonosis density

and all disease density, and then using them as the dependent variable. We have incorporated these important revisions in our Methods section (Line 412-414, Line 427-431). Our new analyses showed that the PubMed citation was indeed one important variable explaining the number of zoonosis events (Line 149, 152) and we have addressed this in our Discussion section (Line 263-266). Importantly, the results based on these new analyses did not change our main finding on the important role of alien zoonotic hosts in promoting zoonosis emergences.

Review Comments 2) *“Regardless, sampling bias should be more prominently discussed in the paper itself rather than just put in the methods, as I spent the time reading the paper wondering if everything was robust to these biases or not.”*

Our Response: Following the reviewer’s comment, we have expanded our discussion about sampling biases to help readers understand this important issue (Line 202-207).

Review Comments 3) *“The analyses could control more effectively for geographic non-independence in their results, rather than merely using country-level metrics. Spatially heterogeneous sampling effort is important in determining observed disease patterns, and some recent analyses have included some very advanced approaches to deal with it. See for example the pubcrawler approach used by Allen et al² and the spatial autocorrelation analyses used by Albery et al³. Fitting smoothers or autocorrelation effects should be attempted, e.g. using the countries’ centroids. Additionally, adding country size as a fixed effect might be informative.”*

Our Response: This is also an excellent comment regarding the potential for geographic biases. We used the PubCrawler method following your suggestion by adding the PubMed citations of each country in our new analysis (Please also see our response to your 1st comment above, Line 412-414). We have also applied the methods of Albery et al.’s (2021) by adding the latitude and longitude of each geographical centroid as predictors in the GAMM fits to control for spatial autocorrelation (Line 414-417). Please note that we did not add country size again because that we have used the total number of zoonosis events divided by country size when we generated the dependent variable to account for the fact that larger

geographical regions may host more disease events (Line 422-427).

Review Comments 4) *“Similarly, is controlling for non-(observed) zoonotic host diversity really sufficient? Given taxonomic and geographic biases in sampling and the fact that the virome is woefully undersampled, is it possible that many of these “non-zoonotic species” just haven’t been observed with zoonoses yet? Even if this measure is used, I appreciate the authors’ use of a literature search to identify zoonotic and non-zoonotic hosts, but there are large databases that can be used to identify these species-level data easily. Most recently, Gibb et al synthesised and published a large meta-dataset that can be used for just this purpose⁴. I incorporated the authors’ zoonotic list from Supplementary Table 2 into Gibb et al.’s data and found an extra >2000 animal hosts of human-infecting parasites that were not mentioned in their original list. I used the definition “were ever seen hosting parasites that have ever been found in humans”, which is a fairly loose definition, but could provide a helpful contrast to their list of 800 or so species. I suggest incorporating this dataset, or if not, incorporating other open datasets of known host-parasite associations.”*

Our Response: We appreciate the reviewer’s careful examination of the data on zoonotic identity of the established alien species. We completely agree with the reviewer’s concern and have carefully re-checked all the host identity of our study species based on the reviewer suggested latest database provided by Gibb et al. (2021). As you will see in our revised manuscript, after incorporating the latest update by Gibb et al. (2021), we now have a total of 93,544 pathogen-host records. We re-conducted all our new analyses with the additional data and obtained similar results. We have incorporated the new database in our revised method section (Line 359-368) and updated our results along with the supporting materials (Supporting Tables 2-3).

Review Comments 5) *“While I appreciate the authors’ desire to do analyses separately by order, I would recommend lumping the orders together, fitting the same variables, and then fitting an interaction effect between order and the effects of interest. This exercise will inform whether orders actually show *different* responses to the explanatory variables of interest, or whether it is parsimonious to keep them together. If the inclusion of the order interaction*

does not help (which I suspect might be the case) this will substantially simplify their reporting, which will help to include the analytical modifications I've outlined above."

Our Response: We are thankful for this excellent suggestion. In the revised manuscript, we have reconducted all analyses by lumping the orders together, with order identity as a random intercept, and find that alien zoonotic host richness is still an important factor that is positively related to the number of zoonosis events. Furthermore, we followed the reviewer's suggestion by including an interaction term between order and alien host richness to test whether there are different responses among host orders. These new analyses were consistent with the original order-separated analyses and there were indeed some orders, such as Carnivora, Cetartiodactyla, Rodentia, waterfowl, Galliformes, Passerine, and Dipteran, that caused a greater increase in zoonosis events than other orders. We have updated all the Methods (Line 476-479) and Results (Line 160-171) in our revised manuscript.

Review Comments 6) *"More generally, these are relatively simplistic generalised linear models, and there are doubtless non-linear relationships in there, so GAMMs would be preferable."*

Our Response: Thank you for pointing out this key issue. In our revised manuscript, we re-conducted all our analyses using GAMMs. Importantly, we obtained similar findings using GAMMs as we showed previously, supporting the robustness of our findings to some model uncertainties. Regarding the GAMMs, we also conducted sensitivity analyses by fitting all variables with thin-plate smooths of 6, 8, and 10 knots. Given that the results were qualitatively similar across all three tested knots, we report results from fitting variables with 10 knots in the main text and provide results from 6 and 8 knots in Supplementary Fig. 3. We have updated all the Methods (Line 431-446, Line 469-475) and Results (Line 138-171) sections in the revised manuscript.

Review Comments 7) *"Showing plots of the relationships between (at least) invasion numbers and EID numbers, including the fit (and 95% intervals) from the models would help to assess the validity of a linear fit, rather than just displaying the forest plot, similar to what they've done with Figure 3. This would also inform other aspects of the data distribution and*

would help to interpret the strength of the relationship and the variation accounted for by the effects. (given that I recommend removing the temporal analysis below, I would replace Figure 3 with the figure I'm suggesting here). Moreover, why have the authors fitted a smoother to Figure 3 if it's a relationship that they tested with linear fits in the main analyses? Why not fit the line from the model?"

Our Response: Thank you very much for this very helpful suggestion. Instead of the previous forest plot, we now provide the scatter plots with and 95% confidence bands between the number of zoonosis events and the important predictor variables from the best model based on AIC, following the reviewer's suggestion (Line 153-155, new Fig. 4, Line 754).

Review Comments 8) *"P values, effect sizes, R², etc. should all be reported more visibly for all models, as should the model formulations in the methods. Having an explicit formula always helps. Most pressingly, although the invasion effect is significant, what proportion of the variance does it account for? These data could be combined with my suggested scatter plots including the model fits to increase their accessibility."*

Our Response: Thank you very much for these suggestions. We are sorry that we did not show the analyses and results more clearly in the original manuscript. In our revised manuscript, we provide a new diagram to show our data analyses, including the model formulation (new Fig. 2, Line 739). We also added related information in the Methods section (Line 465-467), and reported all the key statistical parameters, including the R^2 , deviance explained and the proportion of the variance each predictor variable accounted for in the new Fig. 3 (Line 745).

Review Comments 9) *"36: Can you add mentions of the taxa for which these effects were tested but not observed? Given the removal of important groups, it's unclear how valid these statements are and whether they really belong in the abstract."*

Our Response: We are sorry for this misunderstanding. In our revised manuscript, we have removed all order-specific analyses and updated all results based on our new order-combined analysis based on the reviewer's suggestion (please also see our response to your 5# comment

above). We have described our findings based on new analyses more succinctly in the Abstract (Line 37-43).

Review Comments 10) *“54: needs comma after “aquaculture” or no comma before “such”*

Our Response: We have removed the comma before “such as” (Line 56).

Review Comments 11) *“63-66: I think this passage could do with citing the Park lab’s recent work on biological invasions and the factors driving parasite acquisition⁵ and Carlson et al.’s prediction that climate change-associated range shifts will drive cross-species transmission (and therefore zoonotic risk)⁶.”*

Our Response: We have cited the two recent papers based on your suggestion (Line 66, 72).

Review Comments 12) *“93: “theses” should be “these””*

Our Response: We have revised this sentence based on new analyses (Line 97).

Review Comments 13) *“93: “98: “both of” should be “both”*

Our Response: We have corrected this error in the revised text (Line 109).

Review Comments 14) *“99: Again a good place to cite 5 rather than a review paper.”*

Our Response: We have changed the reference here following the reviewer’s suggestion (Line 110).

Review Comments 15) *“104-106: This order of continents is extremely what would be expected according to geographically heterogeneous sampling bias.”*

Our Response: We have re-conducted all our analyses following the reviewer's suggestion, controlling for the geographic-level bias (Line 412-417). Please also see our response to your 3# and 4# comments.

Review Comments 16) *“107: As is the finding that more occurred in higher latitudes.”*

Our Response: We have tried to address this point regarding the latitudinal pattern in our

revised Results section (Line 155-158). If we did not understand this comment correctly, please tell us how to further address this point. Thanks.

Review Comments 17) *“133: This Keesing and Ostfeld study is a perspective, not a research article, and merely reports the orthodoxy. An Order-level observation like this should be attributed to a research article like the Olival paper or should be reconsidered according to the more thorough analyses by Nardus Mollentze and Daniel Streicker7.”*

Our Response: We have added the two research articles as references based on the reviewer’s suggestion (Line 222).

Review Comments 18) *“135: The fact that these important orders were removed from the analysis casts a bit of doubt on the validity of the findings reported in the abstract, as I mentioned above. How many orders were actually tested versus how many were found to be important? Choosing a correct null (per Mollentze and Streicker7) is very important for making Order-level statements like this.”*

Our Response: Thank you very much for this very constructive comment. In our revised text, we have clarified the general importance of alien zoonotic host in predicting zoonosis events based on the new order-combined analyses in the Abstract section (Line 37-43). Furthermore, we have followed the reviewer’s suggestion by adding an interaction term between order and the alien species richness to test whether there are certain orders that may be more prone to zoonosis emergences than others are. Indeed, our new analyses detect that there are several orders that are particularly more important to explaining the number of zoonosis events (please also see our response to your 5# comment above). We have made these revisions more clearly throughout our revised manuscript (Line 160-171, Line 476-479).

Review Comments 19) *“173: “class” should be “classes””*

Our Response: We have revised this sentence based on our new analysis (Line 258-259).

Review Comments 20) *“190: Showing that two time series are correlated is actually extremely* *fraught*

statistically:https://twitter.com/sethfinnegan1/status/1427358978766360577. This analysis is not informative, and should be removed unless the authors can conduct a more believably unbiased analysis. If the spatial replicate approach can be maintained, that would work better.”

Our Response: Thank you very much for this suggestion and we completely understand your concern with the temporal analyses. In our revised manuscript, we added more details in the Introduction (Line 99-103) and Methods (Line 485-491) sections on the reason why we conducted the temporal analyses to account for the potential issue that there might be a strong spatial correlation that is offset in time. That is, all the disease events in a location might have occurred early in time but all the invasions occurred late in time. If so, then the invasions could not have caused the diseases despite their being a positive spatial correlation. Thus, we collected the temporal data of each disease report and alien animal introductions, and conducted the temporal analysis to rule out this hypothesis. Furthermore, we have also tried to add a breakpoint regression analysis to identify the optimal breakpoint associated with the change of the alien animal invasions and zoonotic disease emergences (Line 506-512). We have updated all the results based on this new analysis (Line 188-195). Finally, we shifted all the results of the temporal analyses to the supplement in response to your concern (Supplementary Figs. 4-5).

Review Comments 21) *“294-297: See above major comment about this table.”*

Our Response: Following the reviewer’s suggestion, we have added Gibb’s recent database and conducted a new round of careful and intensive literature review to validate the host and non-host identity for each alien species used in our study (Line 359-368, please also see our response to your 4# comment above), and have updated the new Supporting Tables 2-3.

Reviewers' Comments:

Reviewer #1:

Remarks to the Author:

I think this version of the manuscript is much improved over the original submission. Whilst I thought the original paper was well-written and addressing an important problem, I thought there were two substantial issues with the analysis, both related to the central problem that whilst the analysis was based on over 10,000 zoonotic events, these were derived from only 161 human pathogens and an unspecified number of alien host species (certainly much less than 10,000). This introduced a substantial issue with lack of independence, which may have been addressed by using GLMM, but it was unclear what the fixed and random effects were.

This new version of the manuscript includes several new components that largely address these original criticisms. First, a new figure 1 uses a bipartite network to better show the relationship between the pathogen taxa and the alien zoonotic host taxa. Second, the analysis is now outlined graphically in figure 2 and the GLMM has been replaced with GAMM analysis (as suggested by the second reviewer – I agree that this is a better way of analysing the data). Now that the substantive issues have been addressed, there remain a few relatively minor issues that I think still need to be dealt with.

Line 100 "highly offset in time". It seems to me (especially given the specific example given) that the issue is not so much an offset in time, but the putative causal agent (the alien animal arrival) occurring after the putative effect (the zoonotic disease arrival).

Line 119 should read "... zoonotic hosts..."

Line 121 - 124. The logic here seems a bit contorted. The argument is made that 35% of established alien animals are zoonotic hosts for more than one zoonosis, and I would expect the evidence for this would be the average number of zoonoses per zoonotic host, but what is then given is the number of zoonotic hosts per zoonosis.

Line 135. This is a minor point, but all of the disease-causing agents are parasites. The term "parasites" on line 135 actually means "parasitic animals", including protozoa and metazoa

Line 166. "Passerine" should read "Passerines", or to be consistent with the Latin names used elsewhere in the paragraph, "Passeriformes".

Line 201. "Criteria" is a plural form, so the sentence should read "... our criteria for zoonotic alien hosts were highly conservative..."

Line 230. "Passerine" again

Figure 1. I think inclusion of this figure greatly helps the argument in the manuscript. I'm assuming that the left-hand column is sorted by the number of aliens zoonotic hosts per pathogen. The right-hand column does not seem to have any logical order – is it merely generated to minimise the number of crossovers? It might be sensible to have it in some form of taxonomic order.

Figure 2

Lines 739-740. Should read "... The role of animal zoonotic hosts..."

In column one, "administrative unite" should read "administrative unit". Overall, I think this figure is a useful way of presenting the analysis schematically

Figure 3. Again, this new figure is very helpful. I am not entirely convinced that using circles with different radii is the best way to present the proportion of deviance explained. It might be better to use the actual percentage as a numeral. Alternatively, one could perhaps consider using stacked bars, although I guess that the various components of the deviance explained are not strictly

additive, which is what stacked bars would imply.

There is also an issue with using GenS in this figure. In the printed version of the manuscript, the figure is likely to be before the methods, and this means that the reader will not have much idea of what GenS represents. Figure 2 has a very brief mention of it, but I think there needs to be something in the results section of the text (remembering that this precedes the methods in the printed or PDF version) to explain what GenS actually is.

Figure 5. Again, I think this is a helpful figure. However, it looks as if the horizontal axis is a bit offset. The first column in each of the panels is alien zoonotic host richness of zero. However, zero on the axis is midway between the column representing zero alien zoonotic hosts and one zoonotic host

Reviewer #2:

Remarks to the Author:

The authors have done an excellent job with their revisions, and the paper is a really great contribution to the literature. Check spelling of "biodiversity" on the x axis in Figure 4; otherwise, this is ready to publish.

Point-by-point response to the reviewers' comments (all changes have been tracked in Word)

Reviewer #1

“ I think this version of the manuscript is much improved over the original submission. Whilst I thought the original paper was well-written and addressing an important problem, I thought there were two substantial issues with the analysis, both related to the central problem that whilst the analysis was based on over 10,000 zoonotic events, these were derived from only 161 human pathogens and an unspecified number of alien host species (certainly much less than 10,000). This introduced a substantial issue with lack of independence, which may have been addressed by using GLMM, but it was unclear what the fixed and random effects were.*

This new version of the manuscript includes several new components that largely address these original criticisms. First, a new figure 1 uses a bipartite network to better show the relationship between the pathogen taxa and the alien zoonotic host taxa. Second, the analysis is now outlined graphically in figure 2 and the GLMM has been replaced with GAMM analysis (as suggested by the second reviewer – I agree that this is a better way of analysing the data). Now that the substantive issues have been addressed, there remain a few relatively minor issues that I think still need to be dealt with.”

Our response: Thank you very much for your positive comments on our revisions and we have made final revisions according to each of your minor issues below.

“ Line 100 "highly offset in time". It seems to me (especially given the specific example given) that the issue is not so much an offset in time, but the putative causal agent (the alien animal arrival) occurring after the putative effect (the zoonotic disease arrival).”*

Our response: We have removed the confusing description on the highly offset in time and made this point clearly according to your suggestion (Line 102).

“Line 119 should read "... zoonotic hosts..."”

Our response: We have changed “zoonotic host” to “zoonotic hosts” (Line 129).

“Line 121 - 124. The logic here seems a bit contorted. The argument is made that 35% of established alien animals are zoonotic hosts for more than one zoonosis, and I would expect the evidence for this would be the average number of zoonoses per zoonotic host, but what is then given is the number of zoonotic hosts per zoonosis.”

Our response: The reviewer makes an excellent point here. In our revised manuscript, we have provided the average number of zoonoses per alien zoonotic host instead of the number of alien zoonotic hosts per zoonosis (Line 133). We have also revised the related descriptions in the Abstract (Line 38-39), Discussion (Line 210) and the

legend of Figure 2 (Line 754, 758-759) regarding this issue.

“Line 135. This is a minor point, but all of the disease-causing agents are parasites. The term “parasites” on line 135 actually means “parasitic animals”, including protozoa and metazoa”

Our response: We have used “parasitic animals” instead of “parasites” in our revised text (Line 145, Line 320-321).

“Line 166. “Passerine” should read “Passerines”, or to be consistent with the Latin names used elsewhere in the paragraph, “Passeriformes”.”

Our response: We have corrected this issue and used “Passeriformes” in the revised text (Line 138, Line 176, Line 240).

“Line 201. “Criteria” is a plural form, so the sentence should read “... our criteria for zoonotic alien hosts were highly conservative...”

Our response: We have corrected this error and used “were” instead of “was” (Line 211).

“Line 230. “Passerine” again”

Our response: Corrected. Please see our response to your 6# comment above.

“Figure 1. I think inclusion of this figure greatly helps the argument in the manuscript. I'm assuming that the left-hand column is sorted by the number of aliens zoonotic hosts per pathogen. The right-hand column does not seem to have any logical order – is it merely generated to minimise the number of crossovers? It might be sensible to have it in some form of taxonomic order.”

Our response: We appreciate the reviewer’s positive comment on this figure and concern on the logical order of the figure column. In order to improve the display effect, just as what the reviewer assumed, we generated the figure based on the default output of the R software to minimize the number of crossovers, which we have clarified in the revised legend of the figure (Line 759-761).

“Figure 2

Lines 739-740. Should read “... The role of animal zoonotic hosts...”

In column one, “administrative unite” should read “administrative unit”. Overall, I think this figure is a useful way of presenting the analysis schematically”

Our response: Thanks for your positive comment on this figure, and we have corrected the two typos in our revised version.

“Figure 3. Again, this new figure is very helpful. I am not entirely convinced that using circles with different radii is the best way to present the proportion of deviance explained. It might be better to use the actual percentage as a numeral. Alternatively, one could perhaps consider using stacked bars, although I guess that the various components of the deviance explained are not strictly additive, which is what stacked

bars would imply.

There is also an issue with using GenS in this figure. In the printed version of the manuscript, the figure is likely to be before the methods, and this means that the reader will not have much idea of what GenS represents. Figure 2 has a very brief mention of it, but I think there needs to be something in the results section of the text (remembering that this precedes the methods in the printed or PDF version) to explain what GenS actually is.”

Our response: Thank you very much for finding this figure helpful. We tend to use the bubble plot because it not only shows the relative proportion of deviance explained by each predictor variable using the same color in one group, but also shows readers which predictor variables are included in the top five models. These are either not easily achieved or well represented by the stacked bars. Following the reviewer’s suggestion, we have addressed the GenS variable in the revised text (Line 84), and provided the full name of GenS in the revised Figure 3. The analysis diagram figure addressing this variable has also been moved to the first figure (Figure 1), which could provide readers the whole analysis schematically and help readers understand each of the predictor variables including GenS.

“Figure 5. Again, I think this is a helpful figure. However, it looks as if the horizontal axis is a bit offset. The first column in each of the panels is alien zoonotic host richness of zero. However, zero on the axis is midway between the column representing zero alien zoonotic hosts and one zoonotic host”

Our response: Thank you again for finding this figure helpful. We apologize for the confusion of the axis representing zero of the alien zoonotic host variable. There is no offset of the horizontal axis because each of the predictor variable was standardized to a mean of zero and standard deviation of one before it was entered into the analyses to better compare the coefficients of different covariates. So, there may be negative values for some variables. There are similar situations in Figure 4. In our original manuscript, we explained this point both in the Method section (Line 478-481) and the legend of Figure 4 (Line 781-782). In our revised text, we further make it clear in the Figure 5 (Line 791-793).

Reviewer #2

“The authors have done an excellent job with their revisions, and the paper is a really great contribution to the literature. Check spelling of “biodiversity” on the x axis in Figure 4; otherwise, this is ready to publish.”

Our response: Thank you very much for the positive comments on our revisions. We have corrected the spelling mistake of “biodiversity” in Figure 4.